# Molecular Mechanisms for Regulating Stomatal Formation across Diverse Plant Species

**DOI:** 10.3390/ijms251910403

**Published:** 2024-09-27

**Authors:** Wenqi Zhou, Jieshan Liu, Wenjin Wang, Yongsheng Li, Zixu Ma, Haijun He, Xiaojuan Wang, Xiaorong Lian, Xiaoyun Dong, Xiaoqiang Zhao, Yuqian Zhou

**Affiliations:** 1Crop Research Institute, Gansu Academy of Agricultural Sciences, Lanzhou 730070, China; zhouwenqi850202@163.com (W.Z.); 18982791166@163.com (J.L.); lys087@163.com (Y.L.); 793867737mzx@gmail.com (Z.M.); hhj007@sina.com (H.H.); wangxj839@sina.com (X.W.); lianxr@126.com (X.L.); dongxy@st.gsau.edu.cn (X.D.); 2State Key Laboratory of Aridland Crop Science, Gansu Agricultural University, Lanzhou 730070, China; 3Ministry of Education Key Laboratory of Cell Activities and Stress Adaptations, School of Life Sciences, Lanzhou University, Lanzhou 730000, China; wwenjin2023@lzu.edu.cn

**Keywords:** stomatal formation, epidermal morphogenesis, plant hormone, environmental factors, signal transduction, gramineous crops

## Abstract

Plant stomata play a crucial role in photosynthesis by regulating transpiration and gas exchange. Meanwhile, environmental cues can also affect the formation of stomata. Stomatal formation, therefore, is optimized for the survival and growth of the plant despite variable environmental conditions. To adapt to environmental conditions, plants open and close stomatal pores and even regulate the number of stomata that develop on the epidermis. There are great differences in the leaf structure and developmental origin of the cell in the leaf between *Arabidopsis* and grass plants. These differences affect the fine regulation of stomatal formation due to different plant species. In this paper, a comprehensive overview of stomatal formation and the molecular networks and genetic mechanisms regulating the polar division and cell fate of stomatal progenitor cells in dicotyledonous plants such as *Arabidopsis* and *Poaceae* plants such as *Oryza sativa* and *Zea mays* is provided. The processes of stomatal formation mediated by plant hormones and environmental factors are summarized, and a model of stomatal formation in plants based on the regulation of multiple signaling pathways is outlined. These results contribute to a better understanding of the mechanisms of stomatal formation and epidermal morphogenesis in plants and provide a valuable theoretical basis and gene resources for improving crop resilience and yield traits.

## 1. Introduction

Plants’ transition from aquatic to terrestrial life was a significant event in the history of Earth’s life. Paleobotanical and Phylogenetic studies have shown that the earliest terrestrial plants originated during the Cambrian and Ordovician periods, 485 million years ago [1]. In adapting to the terrestrial environment, plants had to overcome several critical adverse factors, including difficulty obtaining water and nutrients, damage from UV rays and cosmic radiation, and invasion by new microorganisms [2]. Approximately 410 million years ago, plants evolved a new organ—stomata—to help plants overcome the adverse influences of the terrestrial environment [3]. Plants use stomata to obtain carbon dioxide required for photosynthesis and release vapor for transpiration [4], which help plants quickly adapt to the terrestrial living environment, greatly expanding the plant’s living range. At the same time, plants with well-developed stomata and roots strengthened Earth’s hydrological cycle [5].

Before the appearance of stomata, organisms such as the primitive plants Isoetes used root absorption to obtain the necessary carbon dioxide for photosynthesis [6]. The earliest bryophytes (liverworts) did not have stomata. However, they had small openings on their epidermis with different structures and functions from stomata, facilitating gas exchange with the external environment [7]. Mosses and hornworts were the earliest species to develop stomata. The structure and function of their stomata are similar to those of higher plants, but the formation process is different. Bryophytes directly form guard mother cells (GMC) through a single asymmetric division, and GMCs form stomata through a single symmetric division. The stomatal formation process in bryophytes is not as well defined, and the distribution of stomata needs to be more strictly patterned. For example, in the moss *Funaria hygrometrica*, the symmetric division of GMCs may not be complete, resulting in guard cells (GC) being separated by incomplete cell walls. Stomata may sometimes cluster together in the moss Polytrichastrum formosum [8].

Ferns have a more regulated process of stomatal formation: Before differentiating into GMC, epidermal cells undergo one or two asymmetric divisions to generate subsidiary cells (SC) that facilitate water and ion exchange for the guard cells (GC) [9]. By these amplifying asymmetric divisions, ferns can regulate the density of pores on the epidermis. In *gymnosperms*, subsidiary cells derive from the division of meristemoid or adjacent protodermal cells near the stomata. In *Pinus banksiana* and *Pinus strobus*, meristemoids go through one equal division, forming a GMC and an SC. Subsequently, the SC grows around the GMC or GC in a polar manner, eventually forming a sub-stomatal cavity that surrounds the stomatal opening. At this stage, subsidiary cells show some physiological characteristics of the grass plants [10]. In the dicot *Arabidopsis thaliana*, a higher-model plant, the stomatal formation process is well studied. A subset of epidermal cells with continuous proliferative ability is called mother cell meristem cells (MMC). MMC undergo an asymmetric cell division, forming a large stomatal primordium (SLGC) and a small triangular meristem. The meristemoid cell can go through zero to three amplifying asymmetric divisions of departments to produce a beneficial novel meristemoid cell and multiple SLGCs. In this way, *Arabidopsis* can exponentially lift quantity SLGCs through a single-cell lineage. Eventually, the meristemoid cell differentiates into a GMC. The process involves equal division, resulting in two identical daughter cells. Finally, the inner cell walls of these two daughter cells thicken and separate to form mature kidney-shaped GC. Most SLGCs differentiate into pavement cells, but some SLGCs initiate spacing divisions at sites away from the existing stomata, GMC, or meristemoid cell to form satellite meristemoid cells that continue with another round of stomatal formation. Thus, stomatal formation in *Arabidopsis* strictly adheres to the “one-cell spacing rule” between stomata, meaning at least one epidermal cell must be present to separate the two pores to avoid the formation of stomatal clusters and maximize gas exchange efficiency. In addition, stomata in *Arabidopsis* are distributed in a satellite-like pattern on the leaf surface, and stomata at the exact location can be at different developmental stages [11,12,13].

## 2. Regulation of Stomatal Formation in *Arabidopsis*

### 2.1. Transcription Factors Control Stomatal Cell Fate Transitions

The molecular mechanisms underlying the acquisition of MMC fate by epidermal cells are still unclear. However, the study found that the process requires the basic helix-loop-helix (bHLH) transcription factor SPCH (SPEECHLESS). SPCH has two closely related homologous genes, MUTE and FAMA. SPCH, MUTE, and FAMA are sequentially expressed in stomatal lineage cells and can form heterodimers with the bHLH transcription factors ICE1 (INDUCER OF CBF EXPRESSION 1, also known as SCRM) and SCRM2, respectively, to regulate the cell fate transitions of MMC-M-GMC-GC [14,15,16,17,18,19] (Figure 1).

SPCH is a crucial transcription factor that regulates the initiation of stomatal development [20]. SPCH mutation results in the complete absence of stomata, while the overexpression of SPCH leads to additional asymmetric division in epidermal cells, forming numerous small cells and extra stomata. This demonstrates that SPCH positively regulates the asymmetric division process during stomatal formation and promotes the differentiation of meristemoids (Figure 1). SPCH activates the expression of many stomatal development-specific genes, among which MUTE is a crucial gene. The loss of MUTE leads to continuous asymmetric divisions in the meristemoids without differentiation into guard mother cells (GMCs), forming multiple undifferentiated SLGCs surrounding a meristemoid in the epidermis. The overexpression of MUTE turns all epidermal cells into pores. These results suggest that MUTE negatively regulates the asymmetric division of meristemoids and promotes their differentiation into GMCs [17,18]. Further studies have found that MUTE can activate many DNA and histone methylation genes, implying that MUTE may modify chromatin to alter gene expression and facilitate cell fate transitions (Figure 1). Additionally, MUTE activates different sets of cell cycle proteins. MUTE activates cyclin CYCD5 to promote the equal division of GMCs while simultaneously activating the expression of inhibitors of CYCD5, such as FAMA and FLP (FOUR LIPS), to suppress CYCD5;1 after GMC completes one division, ensuring that GMCs divide only once [20]. Moreover, MUTE also activates the expression of its inhibitor, ERL1, to facilitate fate transition to the next stage after GMC differentiation [21].

The differentiation of GMCs into GCs is driven by FAMA, one of the target genes regulated by MUTE, following the gradual decline in MUTE expression. The loss of FAMA results in continued equal divisions of GMCs without differentiation into GCs, forming a caterpillar-like structure (Figure 1). The overexpression of FAMA causes epidermal cells to differentiate into single guard cells, forming a fish-scale-like epidermal structure [14,15,22]. FAMA negatively regulates GMC division by inhibiting the expression of CDKB1;1, one of the target genes of FAMA. Additionally, FAMA activates the expression of key genes involved in stomatal opening and closing, such as the ion channel proteins KAT1 and KAT2 and the ABA signal transduction components MPK9 and MPK12, endowing the daughter cells with guard cell characteristics. FAMA also interacts with RBR (RETINOBLASTOMA-RELATED PROTEIN) together with FLP, recruiting the PRC2 complex, which preserves the cellular fate of the guard cells by adding H3K27me3 modifications to the stomatal lineage genes to silence the expression of these genes. The MYB family transcription factors FLP and MYB88 function redundantly with FAMA [23]. The loss of FLP leads to additional equal divisions in GMCs, resulting in two adjacent parallel stomata. Although the loss of MYB88 alone does not show a phenotype, the *flp myb88* double mutant exacerbates the FLP phenotype, resulting in the formation of numerous parallel stomata [24]. This shows that FLP and MYB88 jointly negatively regulate the number of isometrical divisions of meristem cells. The target genes regulated by FLP and MYB88 are mainly the components of two classes of cell cycle cyclin complexes, CYCD/CDKA and CYCA/CDKB, both of which promote cell division. FLP and MYB88 inhibit GMC division by negatively regulating the expression of these two classes of transcription factors [24].

Two bHLH IIIb family transcription factors, ICE1 (INDUCER OF CBF EXPRESSION 1) and SCRM2 (SCREAM2), also regulate cell division and differentiation during stomatal formation. The *ice1 scream2* double mutant phenotype resembles the phenotype of *spch*, *ice1 scrm2+/−* shows the phenotype of *mute*, and *ice1* exhibits phenotypic characteristics of *fama*. ICE1 and SCRM2 can interact with SPCH, MUTE, and FAMA. The results show that ICE1 and SCRM2 form a heterodimer with SPCH, MUTE, and FAMA and jointly regulate stomatal formation [14,15,16,17,18]. FLP and SCRM/2 can form heterodimers and also play a key role in the fate of M to GMC (Figure 1) [25]. Further studies have revealed that these transcription factors recruit the RNA polymerase II complex and direct the transcription process of specific target genes [26].

Stomatal formation causes the protodermal cells to acquire the fate of meristemoid mother cells (MMCs: gray). An MMC undergoes an asymmetric entry division and gives rise to two daughter cells, meristemoid (M: yellow) and stomatal-lineage ground cells (SLGC: white). M carries out a limited number of asymmetric amplifying divisions to increase the number of SLGCs while concurrently engaging in self-renewal processes. Finally, M differentiates into the guard mother cell (GMC: blue). Each GMC symmetrically divides to yield a pair of guard cells (GC: green). SLGCs can also acquire the MMC fate and undergo asymmetric division to produce satellite meristemoids that are oriented away from preexisting stomata or precursors. SLGCs terminally differentiate into pavement cells. An SPCH-SCRM bHLH module regulates the initiation of stomatal formation. An MUTE-SCRM-FLP module promotes differentiation of the GMC and coordinates the occurrence of a single symmetric cell division. Meanwhile, an FAMA-SCRM module, along with FLP and MYB88, restricts the execution of this single symmetric cell division, and FAMA promotes GC morphogenesis [24]. The EPFs-ERf-TMM/SERKs/BAK1 ligand-receptor signal regulates SPCH, MUTE, and FAMA, which restricts entry into stomatal cell lineages and enforces proper asymmetric spacing division, respectively. The activated receptor signal is transduced via the YDA-MKK4/5/7/9-MPK3/6 cascade, which suppresses SPCH, probably suppresses MUTE, and promotes FAMA for respective action during stomatal formation. PM stands for plasmalemma. A dashed black arrow indicates that a direct acting signaling molecule has not yet been discovered, and a straight black arrow indicates a promoting or inhibiting effect.

### 2.2. MAPK Signaling Cascade Functions Upstream of Stomatal Transcription Factors to Regulate Stomatal Formation

The mitogen-activated protein kinase (MAPK) signaling cascade is a highly conserved intracellular signal transduction pathway between yeast and humans. The pathway consists of three levels of protein kinases: MAPKKK (MAPK kinase), MAPKK (MAPK kinase), and MAPK, which are activated in turn to transmit and amplify signals. The stomatal signaling pathway is also involved in the MAPK signaling cascade, including the YODA (MAPKKK), MKK4/5/7/9 (MAPKK), and MPK3/6 (MAPK) components. The YODA-MKK4/5/7/9-MPK3/6 signaling cascade negatively regulates SPCH and MUTE [27] (Figure 1). The continual activation of these signaling cascade components can result in the entire epidermis consisting of flat cells. Mutants of *yoda*, *mkk4/5*, or *mpk3/6* show a clustered stomatal phenotype. Further studies have found that there is an MPK Target Domain (MPKTD) on SPCH that is phosphorylated by MPK3/6 to inhibit SPCH activity [28]. MUTE does not possess MPKTD, so MPK3/6 may regulate the activity of MUTE in an indirect way. FAMA also requires MAPK signaling cascade modules for regulation. It has been found that the specific expression of MKK7 using the *FAMA* promoter at the GMC-to-GC differentiation stage leads to the phenotype of clustered stomata and small cell clusters with arrested development. Similarly, the expression of MKK9 using the FAMA promoter also results in the formation of clustered stomata. This indicates that MKK7/9 can both inhibit the cell fate before GMC and promote GC differentiation while also promoting cell division. MKK7/9 positively regulates the maintenance of the stomatal fate by FAMA, but their downstream MPKs and the detailed regulatory mechanisms of this module are not yet clear.

MAPK signaling cascades can be regulated by certain protein phosphatases. A member of the PP2C (phosphatase 2C) protein phosphatase family, AP2C3 (*Arabidopsis* protein phosphatase 2C), is expressed in stomatal cells. The loss of AP2C3 does not cause a stomatal phenotype. However, the ectopic expression of AP2C3 causes almost all cells in the leaf epidermis to transform into stomata, which is similar to the phenotype of MUTE overexpression and mpk3/6 mutants. This suggests that AP2C3 may promote stomatal differentiation through the negative regulation of MPK3/6 [29]. Another negative regulator of the MAPK signaling cascade is MKP1 (MAP KINASE PHOSPHATASE1). The loss of MKP1 leads to the formation of clustered small cell clusters. Genetic analysis indicates that MKP1 is positioned between YODA and MPK3/6 in the signaling pathway, regulating stomatal cell differentiation by suppressing the activity of the MAPK signaling cascade [30]. The majority of these factors work not specifically for stomata but dramatically change the leaf shape and, probably, the auxin distribution following chromatin remodeling. Thus, they may also have an effect on stomata as accompanied events through a hormonal/epigenetic signal in the whole leaf.

### 2.3. Asymmetric Distribution of MAPK Signaling by Scaffold Proteins BASL and POLAR Controls Stomatal Asymmetric Cell Fate

To give daughter cells different fates, plants need to establish cell polarity before undergoing asymmetric divisions. The establishment of cell polarity in stomatal lineage cells is regulated by a key protein called BASL (BREAKING OF ASYMMETRY IN THE STOMATAL LINEAGE). The expression of BASL is regulated by SPCH. When the meristemoid mother cell (MMC) starts expressing SPCH to initiate stomatal lineage formation, BASL also begins to be expressed in the cell nucleus [31,32,33]. Before the MMC undergoes asymmetric divisions, BASL can interact with the negative regulators of SPCH, the YODA-MKK4/5-MPK3/6 signaling cascade, and be phosphorylated. The phosphorylated BASL, along with the YODA-MKK4/5-MPK3/6 signaling cascade, is polarized at the cell periphery. As a result, the nuclear SPCH, without the inhibition of the YODA-MKK4/5-MPK3/6 signaling cascade, starts to activate its own expression in large amounts, initiating asymmetric divisions. After asymmetric divisions are completed, BASL, as a scaffold protein, enriches MAPK signaling around cells and restricts MAPK signaling to SLGCs, thus promoting the phosphorylation and degradation of SPCH and leading to the differentiation of SLGCs into paving cells [32]. But in the meristemoids, lacking the inhibition of the YODA-MKK4/5-MPK3/6 signaling cascade, stomatal formation can continue. If the meristemoid cell undergoes another round of asymmetric divisions, the cycle repeats. Additionally, a polarity protein called POLAR (POLAR LOCALIZATION DURING ASYMMETRIC DIVISION AND REDISTRIBUTION), expressed in the meristemoid stage, is also involved in the establishment of cell polarity and the formation of asymmetric cell division in stomatal lineages. The expression of POLAR depends on BASL [17], and POLAR may recruit BIN2, a key brassinosteroid signaling component, to the cell cortex, alleviating the BIN2-mediated inhibition of BIN2 on SPCH and thus promoting asymmetric cell division in stomata [34].

### 2.4. EPF Ligands and Their Receptors Activate the YDA-MAPK Cascade to Regulate Stomatal Formation

In *Arabidopsis*, there are three main ligands involved in regulating leaf stomatal formation: EPF1 (EPIDERMAL PATTERNING FACTOR 1), EPF2, and EPFL9 (STOMAGEN) [35,36,37]. These ligands are initially expressed as precursor peptides and need to be processed by proteases to form mature ligands. SDD1 (STOMATAL DENSITY AND DISTRIBUTION1) and CRSP (CO_2_ RESPONSE SECRETED PROTEASE) are two types of Bacillus subtilis proteases. CRSP can process EPF2, but SDD1 cannot process EPF1/2. The loss of SDD1 leads to clustered stomata on *Arabidopsis* leaves. However, it is still unknown which ligand SDD1 processes to carry out its function. EPF1 and EPF2 have similar functions, both inhibiting the activity of SPCH and reducing the protein level of SPCH, but they have different expression patterns during stomatal formation. EPF2 is expressed early and mainly regulates the initiation of stomatal development and the proliferation of promeristem cells (MMC) [38,39]. EPF1 is mainly expressed in late meristemoid cells, where it is secreted to the periphery, reduces the level of SPCH in SLGC, and ensures the normal stomatal patterning [38]. Additionally, EPF1 reduces the level of MUTE in late meristemoid cells, preventing additional equal divisions in GMC. EPFL9 is a small peptide secreted by mesophyll cells during leaf development and promotes stomatal formation when more CO_2_ is needed for photosynthesis [39]. In the stem and hypocotyl of *Arabidopsis*, EPFL4 (CLL2), EPFL5 (CLL1), and EPFL6 (CHAL) are secreted by the endodermis, regulating stomatal formation and patterning [40].

EPF1 and EPF2 can activate ERECTA (ER) family proteins, leucine-rich receptor kinases (LRR-RK), including ER and ERL1 (ERECTA-LIKE 1), and ERL2 on the cell membrane to inhibit stomatal formation, while EPFL9 can compete with EPF1/2 for binding to ER, blocking the inhibitory signals transmitted by EPF1/2 and promoting stomatal formation [41]. TMM is a co-receptor of the ER family, and it regulates stomatal development signaling by modulating the binding ability of ERs to different EPFs. In *Arabidopsis* leaves, ERs cannot bind EPFs individually but rely on TMM to bind to specific ligands. For example, ER-TMM specifically binds to EPF2, ERL1-TMM specifically binds to EPF1, and EPFL9 competes with EPF2 to bind to ER-TMM, blocking the signal transmitted by EPF1/2 and promoting stomatal formation [42] (Figure 1). In *Arabidopsis* stems and hypocotyls, EPF4 directly binds to ERL2 to activate the MAPK signaling cascade. The binding of EPF4 to ERL2 is weakened by ER-TMM, thereby shutting down the MAPK signaling cascade and promoting stomatal formation. TMM regulates the ability of EPFs to bind to ERs in a tissue-specific manner, resulting in an aggregated stomatal phenotype in leaves but no stomatal phenotype in stems and hypocotyls.

Apart from TMM, ERs can also interact with another group of co-receptors called SERKs (SOMATIC EMBRYOGENESIS RECEPTOR KINASEs). The SERK family consists of five members named SERK1 to SERK5, which participate in signaling transductions in stomatal formation, immune responses, cell programmed death, and BR signaling pathways. In the stomatal development signaling pathway, SERK3 plays a major role, while other SERKs, such as SERK2, SERK1, and SERK4, have a decreasing order of importance [43]. SERKs can form complexes with ERs/TMM/EPF and activate downstream MAPK signaling cascades (Figure 1). However, their specific roles in the complex are still unclear.

The EPF/ERs/TMM/SERKs signaling cannot directly transmit to the MAPK signaling cascade. There are other components that mediate the signal between these two modules. One such regulator is the VAP-RELATED SUPPRESSORS OF TMM (VST), a class of ER signal modulators that can simultaneously interact with membrane proteins and endoplasmic reticulum proteins, regulating signals by modifying the plasma membrane microstructure. The VST family consists of three members, VST1, VST2, and VST3. Mutations in vst1, vst2, or vst3 lead to the phenotype of clustered stomata and additional cell divisions. ERL2 can interact with VST, and VST activates the downstream MAPK signaling cascade. The loss of VST prevents signals from transmitting to MAPK, resulting in the excessive activation of downstream SPCH components [44].

## 3. Stomatal Formation in Monocot Plants

Stomata in grass plants, members of the Poaceae family, differ significantly from stomata in other plants as they possess typical subsidiary cells (SC). Grass plants can efficiently exchange gases with the external environment through this unique physiological structure, maximize photosynthetic efficiency, and quickly respond to environmental changes. Therefore, grass plants can adapt to various extreme growth environments. Currently, grass plants occupy more than one-third of the land area, and more than two-thirds of the crops are grass plants. These achievements are mainly attributed to their unique stomatal structure [17] (Figure 2). Stomata in grass plants are arranged linearly and strictly adhere to the rule of one-cell spacing. Stomatal formation in grass plants starts from specific cell lineages at the base of the leaf, and as the leaf grows upward, the developmental stages of stomata gradually mature, and mature leaves only have mature stomata without the ability to form new rows of stomata (Figure 2).

The unique structure of stomata in grass plants requires a distinct developmental process to construct them. Some MMCs with proliferative ability in the leaf base generate an SLGC and a meristemoid cell through an asymmetric entry division. This meristemoid cell does not undergo further divisions but directly differentiates into a GMC. The GMC induces its later epidermal cells to adopt the fate of subsidiary mother cells (SMC). SMC undergoes a directed uneven division to generate an elongated SC. Afterward, the GMC undergoes equal division, forming two rod-shaped GCs. Lastly, the elongated SC transitions into an inverted triangle shape, and the rod-shaped GCs elongate into dumbbell shapes, completing the morphogenesis of stomata and developing into mature stomata [45,46].

In grasses, stomatas are arranged in parallel rows flanking veins. The initiation of grass stomata is restricted to the base of longitudinally developing leaves. The proliferating cell files (purple) flanking veins acquire the fate of stomata. The smaller daughter cell undergoes asymmetric division to produce early GMC (orange), which matures (blue) and recruits subsidiary mother cells (SMCs). SMCs cause asymmetric division producing subsidiary cells (SCs: yellow) from lateral neighboring cell files. Finally, GMCs divide symmetrically to form guard cells (GC: green). The GCs and SCs differentiate and become a functional four-cell stomatal complex. During the regulation of grasses, stomatal formation in the *Brachypodium*, maize and rice, and the transcription factors *SPCH*, *SCRM1*, *OsSHR*, *OsSCR*, and *CyclinA2;1* regulates the establishment of stomatal rows and promotes asymmetric division to produce early GMC; *MUTE*, *BdFAMA*, *SCAR/WAVE*, and *PANs/ROPs* promote the differentiation of early GMC into mature GMC and regulate the unequal division of SMC to form SC; *FAMA* and *SCRM2* facilitate the differentiation of early GC into mature GC. The key transcription factors that regulate GMC differentiation into GC have not been identified. Arrows illustrate the process by which proliferating cell files undergo two unequal divisions, resulting in the formation of GMCs and SCs, respectively. Following this, the GMCs undergo an equal division, ultimately culminating in the development of mature stomata. 

This formation process unfolds through a series of asymmetric cell divisions, culminating in the formation of a guard mother cell (GMC) and an accompanying cell that increases in size. Subsequently, the GMC initiates the specialization of adjacent epidermal cells into subsidiary mother cells (SMCs). Asymmetric smooth muscle cell division produces a smaller daughter cell and a larger cell, the latter dedicated to epidermal differentiation. GMC undergoes symmetric division, yielding a pair of guard cells (GCs). Ultimately, through a series of differentiation events, the concerted action of GCs and SCs results in the formation of mature four-cell stomatal complexes, as depicted in Figure 2 [47,48,49].

## 4. Regulation of Stomatal Formation in Grasses

### 4.1. Transcription Factors Control Stomatal Cell Differentiation in Grasses

Through homologous sequence alignment, it was ascertained that SPCH, MUTE, FAMA, ICE1, and SCRM2 showed significant conservation in *Arabidopsis*, *Oryza sativa* and *Brachypodium*, and *Zea mays* [46,47]. Notably, in *Oryza sativa*, the *osspch1* mutant exhibits typical stomatal development, whereas the *osspch2* mutant manifests a reduction in stomatal density. Conversely, the *osspch1-1 osspch2-2* double mutant fails to produce stomata, indicating a collaborative role of OsSPCH1 and OsSPCH2 in orchestrating the initiation of stomatal development in rice, with OsSPCH2 being preponderant [47]. Similarly, in *Brachypodium distachyon*, the *bdspch1* single mutant experiences a minor decline in stomatal density, whereas the *bdspch2* mutant demonstrates significantly reduced stomatal density. Notably, stomata deficiency in the *bdspch1 bdspch2* double mutant underscores the indispensability of at least one form of BdSPCH in priming the stomatal lineage, akin to the functional interplay observed in OsSPCH1 and OsSPCH2 [46].

To minimize water loss and maximize photosynthetic efficiency, the density and the distribution of stomata on the leaf surface should be adjusted according to the growth environment. Plants rely on cell surface receptors and ligands to perceive external signals; they are then transmitted to key transcription factors such as SPCH, MUTE, and FAMA through a mitogen-activated protein kinase signaling cascade to regulate stomatal formation. The *osmute* mutant experiences a developmental arrest of stomatal cells at the GMC (guard mother cell) stage, failing to form SCs. The overexpression of *OsMUTE* promotes the division activity of epidermis, generating numerous small cells. In *Brachypodium distachyon*, the *bdmute* mutant cannot induce SC formation from GMCs, with some GMCs displaying aberrant division orientations or stunted development. Conversely, the overexpression of *BdMUTE* causes epidermal cells to undergo uneven division, generating a substantial number of SCs [46]. In maize, the *bzu2-1*/*zmmute* mutant results in the continuous division of GMCs, producing a bunch of cells that look like a stick of sugar-coated haws, consequently resulting in SC depletion [50].

In rice, the *osfama* mutant showcases some GMCs undergoing equal division to yield two arrested elongated GCs, shedding light on the role of OsFAMA in governing GC differentiation [47]. In *Paspalum*, *bdfama* mutants fail to form functional stomata but instead form a four-cell complex of undifferentiated pairs of guard cells and adjacent accessory cells [51]. The BdFAMA protein exhibits subdued expression in GMCs, peaking before GC division and persisting until GC maturation. At the same time, the *bdmute bdfama* double mutant fails to elicit SC formation, underscoring the redundant functional roles of BdFAMA and BdMUTE in orchestrating the fate transition of GMCs to GCs [51].

The *osice1-2* mutant, characterized by a complete loss-of-function mutation in *OsICE1*, exhibits the complete absence of stomata. Conversely, in the *osice1-1* weak mutant, arrested GMCs and GMCs undergoing continuous divisions akin to those in *mute* mutants were observed. In contrast, the *osice2-1* single mutant displays normal stomatal formation, but the *osice2-1 osice1-1* double mutant fails to produce stomata. The direct interaction between OsICE1/2 and OsSPCH1/2, OsMUTE, and OsFAMA suggests their coordinated regulation across different stages of stomatal formation in plants [47]. Similarly, in *Brachypodium distachyon*, the *bdice1* mutant fails to form stomata. In contrast, the *bdscrm2* mutant presents GMCs resulting from equal division that cease differentiation into mature GCs, underscoring the role of *BdICE1* in initiating stomatal formation and *BdSCRM2* in GC differentiation and maturation [46].

In *Arabidopsis*, the GRAS family member SHR (SHORT ROOT) was initially identified for its ability to translocate from the root vasculature to the endodermis, where it cooperates with SCR (SCARECROW) to orchestrate the development of vascular endodermal and cortical cells [52]. In rice, the expression of *OsSCR1* in stomatal lineage cells suggests their potential involvement in stomatal formation. Mutational analyses demonstrate that *OsSHR* and *OsSCR* are indispensable at various stages of stomatal development in monocots [47]. Additionally, the ectopic expression of maize *ZmSHR1* in rice leads to augmented stomatal density, indicative of its conserved role in promoting stomatal formation [53].

### 4.2. Regulation of Subdidiary Cell Formation

In maize, establishing polarity in the subsidiary mother cell (SMC) is crucial for forming subsidiary cells (SCs). Initially, BRK (BRICK) polarizes at the interface between the GMC and SMC, leading to the subsequent polarization of ZmPAN2 (PANGLOSS 2) and ZmPAN1 [54,55]. This polarization results in the localization of ZmROP2/9 (RHO GTPASE OF PLANTS 2/9) at the contact site between SMC and GMC, activating the SCAR/WAVE (Scar suppressor of cAMP receptor/Wiskott-Aldrich syndrome protein verprolin homologous) complex within the SMC [56]. Subsequently, the SCAR/WAVE complex polarizes at the interface between SMC and GMC, potentially reinforcing the polarity localization of PAN2 at this interface through an unknown protein. The SCAR/WAVE complex further activates the Arp2/3 (Actin-related proteins 2/3) complex, forming actin patches to facilitate nuclear migration towards the GMC; this leads to the asymmetry division of the SMC [57,58].

Recent findings indicate that BdPOLAR exhibits polarity localization at the distal end of the SMC, in contrast to PAN1, thereby regulating the polarity establishment of the SMC [59]. Additionally, in maize, the actin-binding protein WPR (WEB1/PMI2-RELATED) relies on PAN2 for polarity localization at the interface between SMC and GMC, thereby regulating the polarity establishment of the SMC [60]. Mutations in these components result in abnormal polarity establishment in the SMC, leading to defective subsidiary cell formation. Interestingly, in grass species, MUTE has been found to move from the GMC to the SMC, regulating SC formation [47,50]. In maize, ZmMUTE also moves from the GMC to the SMC and binds directly to the *ZmPAN2* promoter to regulate its transcription process, thus impacting subsidiary cell formation [50].

### 4.3. Regulation of Grass Stomatal Formation by Other Components

Studies on EPF/EPFL homologous genes in grass species have elucidated their pivotal role in stomatal formation in rice (*Oryza sativa*), *Brachypodium distachyon*, barley (*Hordeum vulgare*), and wheat (*Triticum aestivum*) [61,62,63,64,65]. In rice, the overexpression of *OsEPF1* and *OsEPF2* drastically reduces stomatal abundance, the stomatal density and index decreased significantly, with numerous stomatal precursor cells arrested at the guard mother cell (GMC) stage [63,66]. Similarly, in rice, the ectopic overexpression of *OsEPF1* or *OsEPF2* significantly reduced stomatal density, while the deletion of *OsEPFL9* leads to reduced stomatal density in rice leaf tissues [61]. In *Brachypodium distachyon*, the exogenous application of BdEPF2 impedes stomatal initiation, an inhibition alleviated dose-dependently by BdSTOMAGEN [65]. In barley, the overexpression of *HvEPF1* hindered stomatal formation and resulted in a significant decrease in stomatal density [64]. Likewise, the overexpression of *TaEPF1* in wheat also leads to a notable decrease in stomatal density [62,64]. Additionally, the interaction between the cell cycle regulatory factors OsCYCA2;1 (A2-TYPE CYCLIN;1) and OsCDKB1;1 (CYCLIN-DEPENDENT KINASE B 1;1) orchestrates initial asymmetric cell division in the rice guard cell lineage [67]. Moreover, *BdYODA1* represses stomatal initiation and influences the differentiation fate of stomatal lineage cells [46]. Recently, it has been found that the bzu3 gene in maize plays a dual role in controlling cell wall synthesis and glycosylation modification by controlling the steady state of udp glc/glcnac during the formation of dumbbell-shaped stomata [68].

## 5. The Effect of Plant Hormones on Stomatal Formation

Plant hormones, including brassinolide, abscisic acid, auxin, and cytokinin, regulate stomatal formation. The BR signaling pathway gene *BIN2* is a target gene of *SPCH*, and *BIN2* can phosphorylate and inhibit *YDA* in cotyledons and directly phosphorylate SPCH in the hypocotyl. Therefore, BR can promote hypocotyl epidermal stomatal formation by inactivating BIN2 and negatively regulate cotyledon epidermal stomatal formation by enhancing the activity of YDA [69]. ABA synthesis defects increase the stomatal density, suggesting that ABA inhibits stomatal formation [70]. During stomatal formation, auxin is involved in the regulation of the transition from nonisogenic to isogenic division [71] and negatively regulates stomatal formation by activating auxin response factor 5 (ARF5) and inhibiting auxin resistance 3 (AXR3) [72]. ARF5 directly inhibits the expression of STOMAGEN in mesophyll cells to suppress stomatal formation [72]; in dark cultured seedlings, AXR3 promotes stomatal formation by acting upstream in the YDA-MAPK signaling cascade. Elevated levels or enhanced cytokinin (CK) signaling promote increased SPCH expression; SPCH directly induces the expression of *Arabidopsis* response regulator 16 (ARR16) and CLAVATA3/EMBRYO SURROUNDING REGION RELATED 9/10 (CLE9/10). ARR16 negatively regulates CK responses, while CLE9/10 inhibits the proliferation of SLGCs and stomata formation. ARR16/17 interacts with CLE9/10 to counteract the proliferative effect of SPCH and jointly determine the composition of epidermal cell types [73]. Local hormone distribution also leads to epigenetic changes and, therefore, regulates stomata.

## 6. The Influence of Environmental Factors on Stomatal Formation

Environmental factors such as light, carbon dioxide concentration, high temperature, osmotic pressure stress, and pathogenic microorganisms also regulate stomatal formation. Light signals have a significant effect on stomatal formation in all species. It has long been known that the number of stomata on the leaves of plants grown in dark conditions is much lower than that on leaves grown in light conditions. Recent studies have revealed the molecular mechanism behind this phenomenon [74]. The blue light receptor cryptochrome (CRY) and the red receptor phytochrome (PHY) can promote stomatal formation. Under blue light conditions, *cry1cry2* mutants have fewer stomata than the wild type, while *phyA* or *phyB* mutants also have a lower stomatal density under red light conditions [75]. Downstream of these two light receptors is an E3 ubiquitin ligase called COP1 (Constitutive photomorphogenesis protein 1). Under dark conditions, COP1 can ubiquitinate ICE1 and degrade it. When CRY and PHY receive blue, red, and far-red light signals, they can inhibit the ubiquitination and degradation of ICE1 by COP1. When COP1 is absent, ICE1 accumulates, causing clusters of stomata to form [76]. The transcription factor PIF4 (phytochrome-interacting factor 4), downstream of *PHY*, also regulates stomatal formation, but its specific mechanism is still unclear [74,77]. Additionally, light can induce mesophyll cells to produce more EPFL9, promoting stomata formation in the epidermis. Red light also promotes the expression of SPCH and B-GATA (GATA transcription factors of the B subfamily). B-GATAs are able to bind directly to *SPCH* promoters and are required for red light-induced SPCH expression [78].

In most species, stomatal density gradually decreases as the concentration of carbon dioxide (CO_2_) increases. HIC (HIGH CARBON DIOXIDE) encodes a wax biosynthetic enzyme in the plant cuticle. HIC mutants have more stomata in high levels of CO_2_ compared to the wild type [79]. β-carbonic anhydrases CA1 and CA4 induce the expression of EPF2 in response to an increased CO_2_ concentration. Additionally, an elevated CO_2_ concentration can induce the expression of CRSP, which processes the EPF2 precursor into its mature form to inhibit the initiation of stomatal formation [80]. Therefore, high levels of CO_2_ may primarily inhibit stomatal formation through the EPF2-mediated adverse regulation pathway.

Due to its ability to perceive light and temperature signals, PHYB can also be influenced by temperature changes, affecting stomatal formation. When the temperature rises, the transcription factor PIF4 accumulates in stomatal cells. PIF4 can directly suppress the transcriptional activity of SPCH, reducing stomatal formation [81,82]. Furthermore, a negative feedback regulation exists between SPCH and PIF4, in which they mutually antagonize each other to determine whether cells develop into stomata. Lower humidity, drought, and stress can activate the YODA-MKK4/5-MPK3/6 signaling cascade, which inhibits stomatal formation by negatively regulating SPCH or MUTE. However, the specific molecular mechanisms underlying these effects remain unclear.

Stomata also serve as entry points for the bacterial invasion of plants [83,84]. Many pathogens can invade plants through stomata; therefore, they secrete effectors that induce plants to produce more stomata, providing entry points for infection. Effectors such as AvrPto, AvrPtoB, and HopA1 have been found to induce clustered stomata. The pathogenic bacterium *Pseudomonas syringae* enters the host plant through stomata and releases the effector protein HOPA1 [72]. The overexpression of HOPA1 in plants can lead to the specific inactivation of MPK3/6, resulting in clustered stomata [69]. Additionally, the expression of the effector proteins AvrPto and AvrPtoB in the plant pathogen Pst (*Pseudomonas syringae pv*. Tomato) also leads to the aggregation of stomata in *Arabidopsis* [43]. AvrPto and AvrPtoB may promote stomatal formation by interfering with the function of their target SERKs. The stomatal clustering phenotype might also be a secondary effect caused by activating immune response core components, such as SERK3 and MPK3/6. Alternatively, effectors may directly promote stomatal formation through other pathways. The exact reasons are still subject to further research and careful validation.

## 7. Summary and Outlook

Stomata play a crucial role in the efficient use of water and the absorption of carbon dioxide in plants. Therefore, it is of great significance to reveal the molecular regulation mechanism of stomatal formation. To date, the molecular and genetic network regulating stomatal formation in the model dicot *Arabidopsis thaliana* has been largely revealed. The core skeleton of this network includes transcription factors, ligand EPFs, the receptors TMM, ERf, and SERKs, the MAPK cascade YDA-MKK4/5/7/9-MPK3/6, and the polarity components BASL and POLAR. Additionally, phytohormones and environmental cues modulate stomatal production mainly through those key signaling components [85]. It is worth noting that a leaf is three-dimensional and contains more layers, including mesophyll and vascular, and that stomatal development is regulated not only by epidermal signals but also by subepidermal and long-distance signals [86]. Moreover, leaf development is regulated by the asymmetric distribution of the phytohormone auxin and the significant divergence of the chromatin status, which is dependent on local hormonal signaling and plays a key role in cell fate determination. Additionally, the expansion level of neighboring mesophyll cells induces mechanical tension on the “stomata” linearity, leading to alterations in the cell morphology and local chromatin status. These changes may serve as the foundation for establishing the stomatal cell fate [87]. Therefore, in the future, building a three-dimensional model of the leaf with cell polarity, a chromatin map, etc. will provide more detailed molecular mechanisms of stomata formation. For example, a recent work showed that MPK3/MPK6 also regulates stomatal formation by suppressing STOMAGEN expression in mesephyll cells [88].

The constitution and morphology of stomata in grass are greatly different from those in *Arabidopsis thaliana*. Moreover, the molecular mechanisms directing stomatal formation in grasses remain poorly understood. However, efforts have been made to improve drought tolerance and yield by the manipulation of stomatal density in crops. For example, decreasing stomatal density by the over-expression of EPF1/2 leads to an increased yield under drought stress in rice and barely, while increasing the stomatal density by mutation *ZmIRX15A* (*Irregular xylem 15A*) results in decreased water use efficiency under drought stress in maize [89]. The identification of novel regulators of stomatal formation in crops in the future will provide more and more potential targets for modulating stomatal production, contributing to crop genetic improvement.

## Figures and Tables

**Figure 1 ijms-25-10403-f001:**
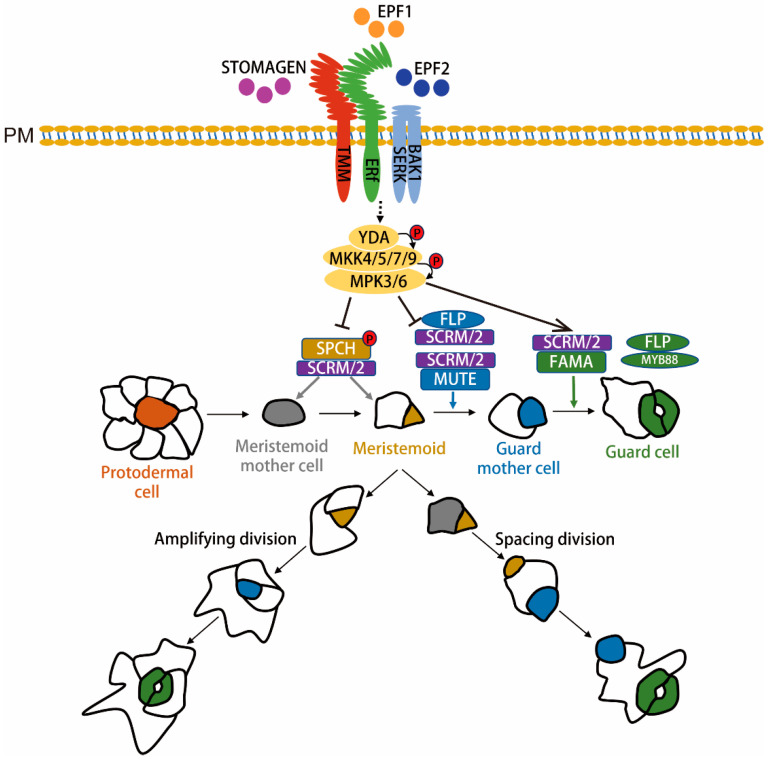
Schematic diagram of stomatal formation and its molecular regulation in dicot plants.

**Figure 2 ijms-25-10403-f002:**
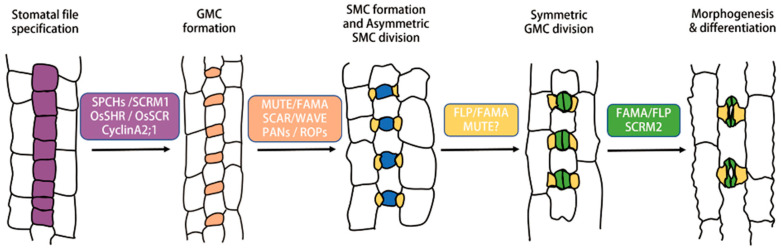
The schematic diagram of stomatal formation in monocot grasses.

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
