# Peer review of "Molecular Mechanisms for Regulating Stomatal Formation across Diverse Plant Species"

_ijms, 2024, doi:10.3390/ijms251910403_

Round 1

Reviewer 1 Report

Comments and Suggestions for Authors

The review is interesting and makes a complete summary of the latests advances on stomatal biogenesis, but I have some suggestions for improvement.

The presentation is very weak. Citations do not follow the IJMS format. Binnomial names are not written in italics (for instance lines 65, 70 and 389). Also the formatting is horrible throughout the text with unnecessary spaces, and lack of them after stops or parentheses. Please revise carefully the format of the manuscript.

The final is very abrupt, without a paragraph of summary or conclusion. Please include a brief conclusion and future prospects at the end of the text.

 I would also include an additional point commenting whether stomatal development has been used for biotechnological improvement of crops. Are there any published crops tolerant to stres or having improved yield thanks to altering the developmental pland and having more or less stomata? This information will greatly increase the interest of the review.

Author Response

Response to reviewer

Reviewer #1:

Comments for the Author:

The review is interesting and makes a complete summary of the latest advances on stomatal biogenesis, but I have some suggestions for improvement.

  1. The presentation is very weak. Citations do not follow the IJMS format. Binnomial names are not written in italics (for instance lines 65, 70 and 389). Also the formatting is horrible throughout the text with unnecessary spaces, and lack of them after stops or parentheses. Please revise carefully the format of the manuscript.

Response:

We first appreciate this reviewer very much for his/her time and the positive evaluation.

Thanks for pointing these errors out. Now, we have revised the format of the manuscript carefully and thoroughly.

  1. The final is very abrupt, without a paragraph of summary or conclusion. Please include a brief conclusion and future prospects at the end of the text. I would also include an additional point commenting whether stomatal development has been used for biotechnological improvement of crops. Are there any published crops tolerant to stress or having improved yield thanks to altering the developmental plant and having more or less stomata? This information will greatly increase the interest of the review.

Response:

We completely agree with this reviewer’s constructive suggestions. We added a brief conclusion and future prospects which includes the information about improvement of drought tolerance of crops through manipulation of stomatal density at the end of the text.

  1. Summary and outlook

Stomata play a crucial role in the efficient use of water and the absorption of carbon dioxide in plants. Therefore, it is of great significance to reveal the molecular regulation mechanism of stomatal formation. To date, the molecular and genetic network regulating stomatal formation in the model dicot Arabidopsis thaliana have been largely revealed. The core skeleton of this network includes transcription factors, ligand EPFs, receptors TMM, ERf and SERKs, MAPK cascade YDA-MKK4/5/7/9-MPK3/6, and polarity components BASL and POLAR. Additionally, phytohormones and environmental cues modulate stomatal production mainly through those key signaling components [86]. It is worth noting that leaf is three-dimensional and contain more layers including mesophyll and vascular, and that stomatal development is regulated not only by epidermal, but also by subepidermal and long-distance signals [87]. Moreover, leaf development is regulated by asymmetric distribution of phytohormone auxin, significant divergence of chromatin status which is dependent from local hormonal signaling and plays a key role in cell fate determination. Additionally, the expansion level of neighboring mesophyll cells induces mechanical tension on the "stomata" linearity, leading to alterations in cell morphology and local chromatin status. These changes may serve as the foundation for establishing stomatal cell fate [88]. Therefore, in the future, building a three-dimensional model of the leaf with cell polarity, chromatin map etc will provide more detailed molecular mechanisms of stomata formation. For example, a recent work showed that MPK3/MPK6 also regulates stomatal formation by suppressing STOMAGEN expression in mesephyll cells [89].

The constitution and morphology of stomata in grass is greatly different from that in Arabidopsis thaliana. Moreover, the molecular mechanisms directing stomatal formation in grasses remain poorly understood. However, efforts have been made to improve drought tolerance and yield by manipulation of stomatal density in crops. For example, decreasing stomatal density by over-expression of EPF1/2 leads to increased yield under drought stress in rice and barely, while increasing stomatal density by mutation ZmIRX15A (Irregular xylem 15A) results in decreased water use efficiency under drought stress in maize [90]. The identification of novel regulators of stomatal formation in crops in future will provide more and more potential targets for modulating stomatal production, contributing to crop genetic improvement.

Reviewer 2 Report

Comments and Suggestions for Authors

The authors made an attempt to provide comprehensive review about stomata induction in Arabidopsis and rice/maize.

Authors definitely collect a lot of publications on the topic, and provide mechanistic view on stomata.

However, authors ignore fact that stomata morphology is a small part of leaf morphogenesis and regulated by signaling coming from other cell type.

Epigenetic changes during stomata formation is completely missing. 

Some details:

Title: very confusing.

You need to separate two process: stomata formation and stomata action/function. It is not clear what do you mean as development here.

Line 12: stomata nothing to do with nutrient absorption (except maybe CO2, what is further convert to carbon through photosynthesis). 

Line 17: “great differences...” is in leaf structure and developmental origin of the cell in leaf, including meristem cell distribution. Stomata just follow these differences.

Line 18: : “ regulation of stomatal development” ???

Lines 91- 289: authors provide nice analysis of TF, AMPKK and other regulator, what, according to authors opinion, “responsible” for stomata formative division.  However if authors carefully examined cited paper, they will find that all these factors work not specifically for stomata, but dramatically changes leaf shape, and, probably, auxin distribution and following chromatin remodeling.. As an example: https://doi.org/10.1105/tpc.109.070110

I would suggest to mention that majority of these factors rather effect on stomata as accompanied events through hormonal/epigenetic signal in whole leaf.

Line 318: “restricted to the base of longitudinally developing leaves” - good statement. Authors need to consider level of expansion of neighbor mesophyll cell which caused mechanical tension on “stomata” linearity, changes cell shape and local chromatin status, what, in turn, is the basis of stomata cell fate establishment.

Figure 1: “development and regulation” ??? here I see only formation.

Figure 2: what mean second row? Is it authors original images?

Both figures represent only schematic 2D drown from one cell layers. However, leaf is 3D and contain more layers (two mesophyll -palisade and spongy in Arabidopsis), vascular etc. Moreover, leaf development regulated by asymmetric distribution of phytohormone auxin and, significant divergence of chromatin status during development. Chromatin status (what is dependent form local hormonal signaling) is a key in cell fate determination. Moreover, authors did not mention role of tension stress in stomata formation.

As conclusion, stomata development need to be evaluated in 3D contents, consider mechanics, genetic and epigenetic, and include local hormonal signaling. Chromatin re-modelling is a key regulator of stomata formation.

Lines 450- 451: this is wrong statement about hormone. Local hormone distribution led to epigenetic changes and therefore, regulated stomata.

Some idea how mechanical tense from inner cell layers can affect epidermis and stomata authors can find here. https://doi.org/10.1093/plphys/kiae408

Lines 474- 491_ the statements are not correct: all mentioned factors and mutants have a multilevel effect, in which stomata is accompanied events.

As conclusion, authors need to be very carefully, not confuse responsibility and accompanied. In the ideal case, authors can build 3D model of the leaf with cell polarity, chromatin map etc to provide detailed “molecular” mechanism of stomata induction.

As extra comments, please check spacing between sentences and reformat literature according to journal rules. 

Comments on the Quality of English Language

in some places spasing missing, some sentences require grammar corrections

Author Response

response to Reviewer #2  

The authors made an attempt to provide comprehensive review about stomata induction in Arabidopsis and rice/maize. Authors definitely collect a lot of publications on the topic, and provide mechanistic view on stomata. However, authors ignore fact that stomata morphology is a small part of leaf morphogenesis and regulated by signaling coming from other cell type. Epigenetic changes during stomata formation is completely missing.

In order to increase impact and readability of this manuscript, I give some improvements below.

Some details:

  1. Title: very confusing.

You need to separate two process: stomata formation and stomata action/function. It is not clear what do you mean as development here.

Response: We first appreciate this reviewer very much for his/her time and the positive evaluation. We completely agree with this reviewer’s constructive suggestions. “formation” is an accurate statement because we focused on stomatal formation in this manuscript. Thus, we use “formation” instead of “development” in the whole manuscript. The title is revised as following: Molecular Mechanisms for Regulating Stomatal Formation Across Diverse Plant Species 

  1. Line 12: stomata nothing to do with nutrient absorption (except maybe CO2, what is further convert to carbon through photosynthesis).

Response:

We completely agree with this reviewer’s comments and have revised the statement as following:

Plant stomata play a crucial role in photosynthesis by regulating transpiration and gas exchange.

  1. Line 17: “great differences...” is in leaf structure and developmental origin of the cell in leaf, including meristem cell distribution. Stomata just follow these differences.

Response:

We completely agree with this reviewer’s comments and have revised the statement as following:

There are great differences in leaf structure and developmental origin of the cell in leaf between Arabidopsis and grass plants.

  1. Line 18: : “ regulation of stomatal development” ???

Response:

We completely agree with this reviewer’s comments and use “formation” instead of “development”. We have revised the statement as following:

These differences affect the fine regulation of stomatal formation due to different plant species.

  1. Lines 91- 289: authors provide nice analysis of TF, AMPKK and other regulator, what, according to authors opinion, “responsible” for stomata formative division. However if authors carefully examined cited paper, they will find that all these factors work not specifically for stomata, but dramatically changes leaf shape, and, probably, auxin distribution and following chromatin remodeling.. As an example: https://doi.org/10.1105/tpc.109.070110

I would suggest mentioning that majority of these factors rather effect on stomata as accompanied events through hormonal/epigenetic signal in whole leaf.

Response:

We completely agree with this reviewer’s comments and added the statement at the end of the section 2.2. The statement is as following:

Majority of these factors work not specifically for stomata, but dramatically changes leaf shape, and, probably, auxin distribution and following chromatin remodeling. Thus, they may also effect on stomata as accompanied events through hormonal/epigenetic signal in whole leaf.

  1. Line 318: “restricted to the base of longitudinally developing leaves” - good statement. Authors need to consider level of expansion of neighbor mesophyll cell which caused mechanical tension on “stomata” linearity, changes cell shape and local chromatin status, what, in turn, is the basis of stomata cell fate establishment.

Response:

We completely agree with this reviewer’s suggestion and added the statement in the final section “Summary and outlook”. The related reference (https://doi.org/10.1093/plphys/kiae408) was cited. The statement is as following:

“The expansion level of neighbor mesophyll cell which caused mechanical tension on “stomata” linearity, changes cell shape and local chromatin status, what, in turn, may be the basis of stomata cell fate establishment (Gloanec et al., 2024).”

  1. Figure 1: “development and regulation” ??? here I see only formation.

Response:

We completely agree with this reviewer’s suggestion and revised the statement as following:

Figure 1. Schematic diagram of stomatal formation and its molecular regulation in dicotyledonous plants.

  1. Figure 2: what mean second row? Is it authors original images?

Both figures represent only schematic 2D drown from one cell layers. However, leaf is 3D and contain more layers (two mesophyll -palisade and spongy in Arabidopsis), vascular etc. Moreover, leaf development regulated by asymmetric distribution of phytohormone auxin and, significant divergence of chromatin status during development. Chromatin status (what is dependent form local hormonal signaling) is a key in cell fate determination. Moreover, authors did not mention role of tension stress in stomata formation.

As conclusion, stomata development needs to be evaluated in 3D contents, consider mechanics, genetic and epigenetic, and include local hormonal signaling. Chromatin re-modelling is a key regulator of stomata formation.

Response:

Yes, the images in the second row in Figure 2 are our original images. We think they are superfluous and deleted in the revised manuscript.

We also completed agree with this review that stomata development need to be evaluated in 3D contents, consider mechanics, genetic and epigenetic, and include local hormonal signaling. This is a very good suggestion and we mentioned it in the final section “Summary and outlook”.

  1. Lines 450- 451: this is wrong statement about hormone. Local hormone distribution led to epigenetic changes and therefore, regulated stomata.

Response:

We also completed agree with this review that Local hormone distribution led to epigenetic changes and therefore, regulated stomata. We added the statement in the section in line 490.

  1. Lines 474- 491_ the statements are not correct: all mentioned factors and mutants have a multilevel effect, in which stomata is accompanied events.

As conclusion, authors need to be very carefully, not confuse responsibility and accompanied. In the ideal case, authors can build 3D model of the leaf with cell polarity, chromatin map etc to provide detailed “molecular” mechanism of stomata induction.

Response:

We completed agree with this review that all mentioned phytohormone and environmental factors and related mutants have a multilevel effect, in which stomata may be accompanied events. This is a very good suggestion and the efforts in the future to build 3D model of the leaf with cell polarity, chromatin map etc will provide more detailed “molecular” mechanism of stomata induction. For example, a recent work showed that MPK3/MPK6 also regulates stomatal formation by suppressing STOMAGEN expression in mesophyll cells (DOI: 10.1093/plcell/koae225). We discussed it in the final section “Summary and outlook”.

  1. As extra comments, please check spacing between sentences and reformat literature according to journal rules.

Comments on the Quality of English Language in some places spacing missing, some sentences require grammar corrections.

Response:

We thank the reviewer for pointing these out. I have done as the reviewer suggested.

Round 2

Reviewer 1 Report

Comments and Suggestions for Authors

Paper has been greatly improved. I can endorse publication.

Congratulations.

Author Response

Thank you again for your time and constructive comments on this manuscript. Have a good life.

Reviewer 2 Report

Comments and Suggestions for Authors

Thank you! The text is almost ready. Small modification of figure 1 still require. Namely, it is not clear what is the upper part/lines means.

My best regards!

Author Response

Thank you! The text is almost ready. Small modification of figure 1 still require. Namely, it is not clear what is the upper part/lines means.

Response: Thank you very much for your thoughtful comments. Upper part/lines in Fig1 represents the cytoplasmic membrane, has to add the PM near the line in Fig1, the original figure illustrates the PM of note is the plasmalemma.